# Cytonemes, Their Formation, Regulation, and Roles in Signaling and Communication in Tumorigenesis

**DOI:** 10.3390/ijms20225641

**Published:** 2019-11-11

**Authors:** Sergio Casas-Tintó, Marta Portela

**Affiliations:** 1Instituto Cajal-CSIC. Av. del Doctor Arce, 37. 28002 Madrid, Spain; 2Department of Biochemistry and Genetics, La Trobe Institute for Molecular Science, La Trobe University, Melbourne, Victoria 3086, Australia

**Keywords:** Cytonemes, *Drosophila*, epithelial cells, Dpp, Hh, EGF, FGF, Wg, glioblastoma, tumourgenesis

## Abstract

Increasing evidence during the past two decades shows that cells interconnect and communicate through cytonemes. These cytoskeleton-driven extensions of specialized membrane territories are involved in cell–cell signaling in development, patterning, and differentiation, but also in the maintenance of adult tissue homeostasis, tissue regeneration, and cancer. Brain tumor cells in glioblastoma extend ultralong membrane protrusions (named tumor microtubes, TMs), which contribute to invasion, proliferation, radioresistance, and tumor progression. Here we review the mechanisms underlying cytoneme formation, regulation, and their roles in cell signaling and communication in epithelial cells and other cell types. Furthermore, we discuss the recent discovery of glial cytonemes in the *Drosophila* glial cells that alter Wingless (Wg)/Frizzled (Fz) signaling between glia and neurons. Research on cytoneme formation, maintenance, and cell signaling mechanisms will help to better understand not only physiological developmental processes and tissue homeostasis but also cancer progression.

## 1. Introduction

Filopodia are long, thin, finger-like, actin-rich plasma-membrane protrusions that function as tentacles for cells to explore their local environment. Cells develop filopodia in response to chemo attractive signals in the microenvironment. These structures are 0.1–0.3 µm in diameter and contain parallel bundles of 10–30 actin filaments held together by actin-binding proteins, including tropomyosin and fascin, and their elongation is mediated by formins [1]. Filopodia dynamics are mainly regulated by the small Rho GTPase, Cdc42 [2]. Filopodia sense the extracellular environment at their tips using cell surface receptors, and they have been given different names usually according to their size or functions: microspikes [3], thin filopodia [4], thick filopodia [5], gliopodia [6], myopodia [7], growth cone filopodia, and dendritic spines involved in synapse formation [8] and in neuronal targeting and pathfinding [9], invadopodia (invasion) [10], podosomes (cell adhesion) [11,12], antigen presentation by dendritic cells of the immune system [13], telopodes [14], pseudopods, tunneling nanotubes [15], and cytonemes [16]. Other related functions include: force generation by macrophages [17], virus transmission [18], vasculogenesis [19], wound closure [20], dorsal closure during *Drosophila* embryonic development [21], Delta-Notch signaling [22], and growth factor signaling [16].

A cytoneme is defined as specialized types of signaling filopodia that exchange signaling proteins between cells. They were first noted as long cellular extensions that protrude from *Drosophila* wing imaginal disc cells, and are predominantly linear, with diameters estimated at 100–200 nm and lengths between 2–150 μm [16,23,24,25]. However, filopodia can extend more than 800 μm and have been measured with diameters of 100–500 nm [16,26,27]. Therefore, the macrostructural features contribute to the classification of the various types of filopodia-like structures.

In the vinegar fly, *Drosophila melanogaster*, cytonemes were initially found in wing and eye imaginal discs [16] and later in ovaries [28], trachea [29,30], and lymph glands [31]. They have also been described in other organisms, such as earthworms [32], earwig ovaries [33], spider embryos [34], in *Rhodnius* and *Calpodes* [35], and in several mammalian cell types including retroviral-infected cells [36], mast cells [37], B-lymphocytes [38], and neutrophils [39]. Recent observations suggest that cytonemes also have an important role during development of the zebrafish neural plate [40], where they transport Wnt8a between distant epithelial cells during development in chicken embryos where they mediate Wnt signaling [41,42], and of the chick limb where they transport Sonic Hedgehog (Shh) [25].

Studies in *Drosophila* have shown differences among cytoneme subtypes. There are cytonemes that send and others that receive signaling proteins. Moreover, cytonemes involved in Decapentaplegic (Dpp), Hedgehog (Hh), Epidermal growth factor (EGF), and Fibroblast growth factor (FGF) signaling can be distinguished by composition, location, and behavior (reviewed in [43]).

Here we will review the physiological role of cytonemes during development in different tissues and the role of cytonemes in tumorigenesis, focusing mainly on the *Drosophila* model organism.

## 2. Cytonemes in the Physiology of Epithelial Cells in *Drosophila*

In this section, we will review and summarize the available information about the structure and composition of cytonemes, available markers, and components required for cytoneme formation. Most of the available literature is based in *Drosophila melanogaster*. Later, we will review cytoneme physiological roles in cell–cell signaling in both epithelial and in non-epithelial cells.

### 2.1. Cytonemes: Structure, Composition, and Markers

The cytoneme core is composed of actin filaments that can be marked with actin fluorescent protein chimeras [23,30,44] and with actin-binding fluorescent protein chimeras, such as moesin (GMA:GFP) [23,45,46] and Diaphanous (Dia:GFP [47]). Dia is a member of the formin family [48], which are involved in actin polymerization and that associate with the growing end of actin filaments. Cytonemes can also be observed by using fluorescent tags to mark either the cytoneme membrane (e.g., CD8:GFP, CD8:Cherry, or Cherry-CAAX), or cytoneme components (e.g., signaling protein receptors such as Tkv [24,44], Breathless (Btl) [24,30], Patched (Ptc) [49], and Fibroblast Growth Factor Receptor (FGFR) [50]. Other components include signaling proteins such as Dpp [24] and Hh [23], and the following components of signaling pathways: Ihog, Brother of Ihog (Boi), Shifted (Shf/DmWif), Dallylike (Dlp), Dispatched (Disp) [23,51], Delta (Dl) [22], and cell adhesion proteins, including Neuroglian (Nrg:GFP), and Capricious (Caps:GFP)) [24,44,52]. Fluorescently labeled Flotillin-2 (Flo2/Reggie-1), CD4-Tomato, and glycosylphosphatidyl-inositol (GPI-YFP) also mark cytonemes [23].

Cytonemes can have different components within the same tissue. For instance, the wing disc’s apical cytonemes contain components of the Dpp pathway (Tkv) [44], the basal cytonemes contain components of the Hh pathway (Hh, Ihog, Dally, Dlp, Shf/DmWif, a secreted protein that positively modulates Hedgehog signaling, Disp, and Ptc) [23,49,51,53], and cytonemes in the air sac primordium (ASP) of the wing disc contain either Btl or Tkv, but not both [24]. It is possible that every signaling pathway has a dedicated set of cytonemes that mediate trafficking between specific signaling cells.

The appearance and physical characteristics of cytoneme tips suggest that they are specialized regions. The tips of ASP cytonemes concentrate over-expressed Nrg:GFP, activated Dia:GFP, indicating that cytoneme tips may be sites of actin nucleation, and Caps:GFP, and the only cytonemes that take up Dpp are those whose tips contact wing disc cells [43]. Many wing disc cytonemes had bright bulbous tips at apparent points of contact with ASP cells [52]. The shafts of cytonemes are marked with either membrane-tethered fluorescent proteins or constituent protein fluorescent constructs. They have a uniform diameter, visualized with fluorescence optics, and cytoneme tips are brighter and wider.

### 2.2. Regulatory Mechanisms in the Formation of Cytonemes

In epithelial tissues, cytonemes emanate from specific membrane territories that have an intrinsic basolateral polarization. Furthermore, in *Drosophila*, the signaling ligands Hh, Wg, Delta, and Spz, as well as their reception processes show basolateral positioning [22,49,54,55,56]. This subcellular localization implicates mechanisms that drive both signaling components and machinery for the initiation of protrusion to the basolateral side. In the *Drosophila* wing disc, an apico-basal activity gradient of the RhoGTPase, Rac, regulates filopodial polarization [57,58]. This Rac gradient is regulated by adherens-junction (AJ) proteins and drive both the position and shape of epithelial filopodia. Likewise, a vesicle-sorting mechanism has been described to transport signaling ligands to the basolateral side [59,60]. However, the regulatory mechanisms for cytoneme cargo upload have yet to be determined, and it is also unknown whether or not a vesicle-recycling mechanism could also contribute to cytoneme formation. In zebrafish, Wnt8a at the plasma membrane recruits transducer of CDC42-dependent assembly protein 1 (Toca-1), which locally activates cytoneme nucleation [40,61]. Hence, intracellular trafficking of the Wnt ligand could be key for the spatial localization of membrane protrusion and signaling.

In *Drosophila*, ectopic expression of a constitutively active form of *Dia* [62] concentrates at the tips of the ASP cytonemes, and cytonemes do not extend normally in the absence of Dia function [52]. The Rho family member, Vav, localizes to wing disc basal cytonemes [44]. Additionally, the *Drosophila* capping proteins SCAR and pico, which are actin-binding proteins, have been implicated in cytoneme function by genetic loss-of-function studies [23].

The ASP cells that extend the cytonemes containing receptor-bound Dpp are able to activate Dpp signal transduction. ASP cells that are genetically compromised for *Dia* and *shibire* (which encodes a dynamin) and *Nrg* or *Caps* (which encodes cell adhesion proteins), fail to make normal cytonemes and are signaling deficient [63]. Moreover, genetic conditions that deplete anterior cells of either *Dia* or *SCAR*, reduce the length and number of cytonemes and reduce both the Hh gradient and signaling in the anterior compartment of the wing disc. Furthermore, over-expression of *Flotillin-2*, a major component of membrane microdomains, increases cytoneme length and the extent of the Hh signaling domain, and it is able to induce numerous filopodia-like protrusions in various cell lines [64].

### 2.3. Signaling and Communication through Cytonemes

Cytonemes have been described to be associated with components of a specific signaling pathway, even when emerging from the same cells [24]. They can transport either the pathway ligand or the receptor, depending on whether they emanate from receiving or signal-producing cells, or from both. Cytonemes have been shown to be involved in the paracrine transport of the following signaling molecules, including Notch, Spi/EGF, Branchless (Bnl)/FGF, Dpp/BMP, Wingless (Wg)/WNT, and Hh/Shh. Cytoneme-mediated delivery of signaling ligands has been shown for Dpp in the *Drosophila* wing disc, Wnt in zebrafish [40,65], Hh in *Drosophila*, and Shh in chick limb bud [23,25,49,55,59]. Conversely, cytonemes emanating from signal-receiving cells have been shown to participate in the distribution of the *Drosophila* FGF receptor homolog Breathless (Btl) in the developing ASP [66].

Establishing the Dpp gradient in the *Drosophila* wing disc depends on cytonemes containing the receptor Tkv. These cytonemes extend from cells situated on both sides of the source territory containing Dpp from producer cells [16]. Cells that activate Dpp signal transduction extend cytonemes to the closest cells that produce Dpp. This specificity is evident in the wing disc, where cells in the wing blade primordium direct Tkv-containing cytonemes toward the Dpp-producing cells at the disc midline to which they are closest. Specificity is also evident in the ASP cells that direct Tkv-containing cytonemes toward nearby Dpp-producing cells of the wing disc, while also directing FGFR-containing cytonemes toward wing disc cells that express FGF. Furthermore, cells in the eye disc direct the epidermal growth factor receptor (EGFR)-containing cytonemes toward Spi/EGF-producing cells of the morphogenetic furrow (MF). However, experimental conditions that change the location of the signaling cells also change cytoneme distributions. Changes to the location of cells that express signaling proteins are reflected in altered distributions of cytonemes and blocking signal transduction, either after ectopic expression of a dominant-negative receptor or if expression of the signaling protein is reduced or eliminated, results in the absence of cytonemes [63].

Cytonemes orient uniformly towards the anterior–posterior (A/P) compartment boundary of the *Drosophila* wing pouch primordium [16]. Cells at the anterior–posterior (A/P) compartment boundary express Dpp (a member of the transforming growth factor-β (TGF-β) superfamily), and long filopodia that extend from wing disc receiving cells towards Dpp-expressing cells at the compartment border are involved in Dpp signaling [16,44]. Different mechanisms were proposed to mediate extracellular molecule distribution among epithelial cells, including extracellular diffusion [67], cell to cell transfer [68], vesicle transport [69], or cytoneme mediated movement [16]. The contribution of cytonemes to Dpp distribution seems to be the most accepted mechanism in the case of the tracheal system. This organization suggested that physical contacts arise at which Dpp transfers to its targets, as an alternative to the diffusion–secretion established model. This mechanism of direct delivery is similar to neurotransmitter release and uptake.

The Hh morphogen is required during development, and Hh signaling has been related to axon guidance, cell migration, stem cell maintenance, and oncogenesis [70]. The role of Hh as a cytoneme based signaling molecule, was originally described in *Drosophila* wing epithelial cells and in abdominal histoblasts [23]. Actin-based cytonemes are produced and transported by Hh-producing cells to deliver Hh to several cell diameter distances. This study showed that Hh gradient correlates with cytonemes formation, and mutations affecting cytoneme formation also disrupt the Hh gradient [23].

There are cytonemes that are specific to the *Drosophila* eye and wing discs or to tracheal cells. Eye differentiation in *Drosophila* is marked by a wave of cell division and differentiation, termed the morphogenetic furrow (MF), which initiates from the posterior margin of the eye imaginal disc just prior to metamorphosis. The MF progresses through the unpatterned dividing cells of the eye disc from posterior to anterior, leaving behind ordered cell clusters called ommatidia [71,72,73]. In the eye disc, cytonemes on the apical surface of columnar epithelial cells orient to either the MF or the equator, which is orthogonal to the MF and defines a line of mirror-image symmetry where dorsal and ventral ommatidia are juxtaposed [74]. EGFR is present in motile puncta in the cytonemes that orient to the MF, where the EGFR ligand, Spi/EGF, is produced [24]. In the wing disc, Thickveins (Tkv), a receptor for Dpp, is present in motile puncta in cytonemes that orient to the disc midline, where *Dpp* is expressed, and the cytoneme tips contact and appear to be directed only to Dpp-expressing cells, suggesting that they transport Dpp across the disc [44]. In the eye disc, cells extend two types of cytonemes, either orienting toward the MF or toward the equator (Figure 1).

The EGFR signaling pathway sustains multiple functions during eye development, including proliferation and differentiation [75,76,77,78]. In normal conditions, cells extend cytonemes to the usual signaling centers [63]. The EGF receptor concentrates in motile puncta in the MF-directed cytonemes but is not present in the cytonemes directed to the equator. When a dominant-negative EGFR is ectopically expressed throughout the eye disc, the long MF-directed cytonemes are absent, suggesting that they depend on Spi/EGF signaling [24]. Conversely, when the EGF ligand is uniformly over-expressed, the only eye disc cytonemes detected are short and lack a directional bias, suggesting that these cytonemes may have orientations and lengths that are dependent specifically on the source of spi/EGF [24,63]. These characteristics suggest that cytonemes in the eye disc selectively localize EGFR; the MF directed cytonemes mediate EGF signaling; cytonemes appear to link signal producing and receiving cells; and stable contacts via cytonemes require contributions from both signal-producing and signal-receiving cells.

In *Drosophila*, cytonemes containing the Fz receptor from the ASP cells contact the Wg-producing cell bodies in the wing disc epithelia for signal reception [54]. In the zebrafish embryo, the tips of cytonemes from Wnt8a-producing cells transfer the ligand by contacting the cell body of responding cells [40]. Moreover, cytoneme distributions and plasticity reveal their specificity for a specific signaling protein. Eye disc cytonemes change distribution after uniform overexpression of *Spi/EGF* and orient towards these ectopic sources, but they do not change after uniform overexpression of *Dpp* or *Hh*. Similarly, ASP cytonemes change after ectopic expression of *FGF*, and wing disc cytonemes change after ectopic expression of *Dpp*, but these cytonemes do not change after uniform overexpression of *Spi/EGF* or *Hh*. These behaviors suggest that stable contacts require contributions from both the signal-producing and signal-receiving cells.

The presence of the Tkv, Btl, and EGF receptor in different, specific cytonemes suggests that each of these cytoneme subtypes mediates the movement of Dpp, FGF, and Spi/EGF, respectively. Cells that activate the signal transduction pathways for these signaling proteins also extend cytonemes that contain the cognate receptor. Similarly, wing disc epithelial cells that activate Dpp signal transduction extend Tkv-containing cytonemes toward Dpp-expressing cells. Likewise, cells of the eye disc activate EGF signal transduction and extend EGFR-containing cytonemes towards the furrow. Moreover, the cells at the tip of the ASP that activate the FGF signal transduction, extend FGFR-containing cytonemes, and cells that activate Dpp signal transduction, extend Tkv-containing cytonemes [63].

Studies in several other systems have reported that signal transduction is associated with cell–cell contacts both for cells that are far apart and for cells separated by short distances. For example, Hh is involved in juxtacrine signaling in the *Drosophila* germline stem cell niche, and it is localized in cytonemes that extend from Hh-expressing cells [28]. In the *Drosophila* leg mechanosensory organ, Spi/EGF is produced in a socket cell and induces a specific neighbor to adopt a bract cell fate. Polarized protrusions that originate from the socket cell appear to target EGF signaling to the particular precursor cell [79]. Filopodia-mediated contacts between cells that are not immediate neighbors have also been implicated in Notch and Scabrous-dependent signaling that pattern the bristles of the adult thorax [22,80]. Cytonemes have been shown to link wing disc and ASP cells and are required for signaling. Dpp in transit between the wing disc and the ASP colocalizes with the Tkv receptor in puncta at cytoneme contacts, also known as cytoneme synapses due to their similarities to neuronal synapses [63], and moves along them [52]. Moreover, contacts at cytonemes from myoblasts carrying the Notch ligand, Delta (Dl), also contact the ASP, and vice versa, suggesting a possible cytoneme–cytoneme interaction to activate Notch signaling [54]. In the case of Hh distribution within *Drosophila* epithelia, a study revealed direct cytoneme–cytoneme contact sites between distant producer and receptor cells all along their length [55]. Similar contacts have been suggested for Shh signaling in the developing chick limb bud [25].

## 3. Cytonemes in Nonepithelial Tissues: Trachea, Myoblast, Ovary, Brain

Various laboratories described the existence and function of cytonemes in the trachea, myoblast, ovary, and brain cells of *Drosophila*. The functions of filopodia in neurons are related to path finding [9] and synapse formation [8], chemotaxis in *Dictyostelium discoideum* [81], cell migration and adhesion [82,83], cell signaling [22,40,52,55,84], and cancer progression and metastasis [83,85,86,87].

### 3.1. Trachea

The tracheal air sac primordium is a branch that interacts with the epithelial cells of the wing disc. Cytoneme extend from the basal surface of the tracheal epithelium and mediate Dpp signaling through its receptor Tkv [44] or FGF signaling through the FGF-receptor [30]. However, these two receptors are not present in the same cytoneme [52], which suggests a higher level of specificity in cytoneme-mediated signaling.

FGF mediates the budding of air sac precursors and tracheal branching during third instar larvae. These air sac precursors extend cytonemes towards FGF expressing cells and establish filopodial contacts. As a result, FGF signaling induces mitosis on differentiated tracheal cells [30].

A recent publication investigating the similarities between neuronal and cytoneme synapses, studied the roles of neuronal synapses components in the development of the *Drosophila* ASP. Signaling in the ASP was disrupted if genes associated to glutamate signaling were silenced in wing disc cells, thus cytoneme-mediated signaling in epithelial cells is glutamatergic. In more detail, partial loss-of-function conditions in the wing disc, which targeted essential components of presynaptic neuronal compartments, decreased the presence of cytonemes and signaling in the ASP, but targeting these genes in the ASP had no effect. Partial loss-of-function conditions in the ASP that targeted essential components of postsynaptic neuronal compartments, decreased signaling in the ASP, but targeting these genes in the wing disc had no effect on the ASP. This indicates glutamatergic functions of neuronal presynaptic compartments only in the signal transmitting cells of the wing disc, and glutamatergic functions of neuronal postsynaptic compartments only in the signal-receiving cells of the ASP [88]. Cytoneme contacts are characterized by GRASP fluorescence, a technique that marks sites of approximately 20–40 nm cell–cell apposition originally developed to identify neuronal synapses [52,55]. Cytoneme contacts and cytoneme mediated signaling depend on the adhesion proteins Caps and Nrg, which also have essential trans-synaptic roles in neuronal synapses [89,90]. This study provides evidence for additional components that are common to both cytoneme contacts and neuronal synapses, including voltage-activated glutamate transmission. It was also found that Dpp signaling in the ASP was compromised if disc cells lacked Synaptobrevin and Synaptotagmin-1 (which function in vesicle transport at neuronal synapses), the glutamate transporter, and a voltage-gated calcium channel, or if ASP cells lacked Synaptotagmin-4 or the glutamate receptor GluRII [88].

### 3.2. Myoblasts

Myoblasts are the precursor cells of the muscle fibers during development; these cells develop together in the wing imaginal disc and later differentiate into the flight muscles. Long-range signals among cells assure the proper regulation of embryonic development. Cytonemes connect myoblasts with epithelial cells and the air sac primordium. Specific cytonemes from myoblast accumulate the Fz Receptor, which takes Wg from epithelial cells. Additionally, myoblast specific Delta (Dl) containing cytonemes contribute to Notch activation in the air sac primordium [54] (Figure 2).

### 3.3. Ovary/Testis

Stem cell niches maintain the proliferative condition of stem cells, but not of the neighboring daughter cells. Niche signals are limited to the stem cells through microtube-based nanotubes that mediate Dpp signaling. Nanotubes accumulate the Tkv receptor that, after interaction with the ligand Dpp, activates signaling within germ-line stem cells. This signaling (Tkv-Dpp) is sufficient to stimulate nanotube formation. Disruption of this signal leads to germline stem cell loss [91]. Microtube-based nanotubes can transport signaling molecules in a similar way compared to cytonemes, but are microtubule-based and F-actin independent [92].

The maintenance of the stem cell niche in the female germline depends on the Hh gradient and cytoneme formation [28]. In this case, Hh is produced in the cap cells of the ovary niche under the transcriptional regulation of *Engrailed*. Hh is secreted to the adjacent escort cells and stimulates the expression of *Dpp* and another Tkv ligand, Glass bottom boat (Gbb), by suppressing Janus kinase signal transducer (JAK/STAT) activity [93]. Whether escort cells are the functional and unique source of Dpp is a matter of controversy. It is also proposed that Dpp is produced by cap cells and functions over a short (one cell diameter) distance to promote GSC self-renewal, by suppressing the expression of the differentiation-promoting factor *bags of marbles* (*Bam*) [93]. Cytonemes are produced in the stem cell niche cap cells and mediate Hh transport and delivery to the escort cells. These cytonemes project directionally towards low Hh signaling in the niche even though the signals that modulate cytoneme length from the niche are still under debate. This system ensures the production of Dpp and Gbb to activate Dpp pathway in female germ-line stem cells to assure their proliferative condition [28].

### 3.4. Neurons

The nervous system cells, neurons, are morphologically distant from other cell types because they extend long protrusions (axons and dendrites) and are polarized and asymmetric. Neurons communicate with target cells or tissues by extending dendrites and axons that form specialized contacts (synapses). These synapses can release or take signals from other cells and ensure the specificity of the communication, and modulate the duration and amplitude of the message [94]. A main characteristic of neurons is the formation of a network that interconnects thousands of cells in the brain. This network facilitates the coordination among different brain regions, contributes to the exchange of molecules (neurotransmitters), and the localization of organelles such as mitochondria in specific zones [95]. These projections are named tunneling nanotubes (TNTs) and share some features with cytonemes [96]. TNTs are formed by F-actin TNTs. In addition, TNTs also contain microtubules, depending on the size and the corresponding delivering cargo. Among others, TNTs transport prion aggregates involved in neurodegenerative diseases and have been postulated as a source of neurodegeneration spreading through different areas of the brain [92].

### 3.5. Glia

Glial cells (microglia, oligodendrocyte, and astrocytes) are specialized nervous system cells that give metabolic support to the neurons and are responsible for the immune response in the brain, contribute to the spreading of the electric signals, recapture glutamate, and are involved in many other essential functions for the brain physiology [97]. Glial cells play a central role in neural diseases such as Alzheimer’s disease [98], amyotrophic lateral sclerosis (ALS) [99], or psychosis [100].

The coordination between glial cells and neurons is essential for axonal conduction, synaptic transmission, and information processing during development and during adult life; therefore, it is critical for brain function. Different methods of communication have been described in the glia–neuron two-way communication, including ion fluxes, neurotransmitters, cell adhesion molecules, extracellular vesicles, and signaling molecules [101,102]. Thus, glial cells have emerged as central players in the development and function of complex nervous systems from flies to humans [103].

In addition, a novel mechanism of cell to cell communication based on cellular protrusions (filopodia) has been described in glial cells [104]. Astrocytes in vivo extend thin processes around synapses that mediate the communication with neurons. These structures are known as peripheral astrocyte processes (PAPs) and are from 50 to 100 nm thick [104].

## 4. Cytonemes in Pathology: Tumorigenesis

The nature of these interconnecting structures and their similarities with epithelial cytonemes are currently under debate. Cytonemes have been proposed to mediate communication between neoplastic cells and cells in their microenvironment [87]. In a *Drosophila* wing disc tumor model utilizing ectopic expression of the *EGFR* and receptor protein-tyrosine kinase (*Ret*) oncogenes, cytoneme formation is required to receive signals from the neighboring cells. Genetic ablation of cytonemes prevents tumor progression, restores apico-basal polarity, and improves survival [87]. This recently established system serves as an optimal platform for novel pharmaceutical approaches against cancer progression in vivo. The authors identified pharmacological combinations against cytoneme-mediated oncogenic signals that prevent tumor progression and improve life span. The value of flies as a valid platform for human disease has accumulated evidences that favor this model for future preclinical studies. In particular, the high cost of testing single or combined pharmacological treatments in mice is several orders of magnitude higher than *Drosophila* based platforms, which has made preclinical trials risky and challenging. The molecular basis underlying cytonemes, the signals transduced by cytonemes, and the implications in tumorigenesis are hot topics for human disease that open novel avenues for potential future treatments.

In addition, the discovery of tunneling nanotubes (TNTs) brings a novel class of thin and long membrane protrusions that connect benign tumor cells [15]. These protrusions form complex networks that mediate the selective transfer of vesicles, organelles, and small molecules [105,106]. TNTs are a common phenomenon in different cell types and tissues that increase under pathological conditions, such as infections, cancer, or neurodegenerative diseases [105]. One considerable limitation to the study of TNTs is the fragility of these structures that makes TNTs difficult to preserve after fixation of tissues. This brought a controversy about their existence in vivo. However, intravital techniques enabled the study of TNTs in live animals [107], which revealed that TNTs are indeed relevant cellular structures in vivo.

In vivo microscopy methods have been used in recent years to study in detail cellular features of cancer cells. A recent study showed that TNTs are induced by stress in prostate cancer and they had a role in mediating intercellular communication that confer stress adaptive cell survival and treatment resistance on the tumoral cells [108]. Additionally, pancreatic cancer cells show TNTs and their formation is stimulated after chemotherapy exposure [109]. Furthermore, TNTs are involved in the communication between tumor cells and macrophages to promote macrophage-dependent tumor cell invasion both in vitro and in an in vivo zebrafish model [110]. Interestingly, colorectal cancer cells have the ability to form locomotory and invasive filopodia that promote invasion and metastasis, and this is suppressed by the phosphorylation of Vasodilator-Stimulated Phosphoprotein (VASP) [111]. Related to colorectal cancer, leucine-rich-repeat containing G-protein-coupled receptor 5 (Lgr5), which labels crypt stem cells, represents the cell of origin in gastrointestinal cancers [112], and Lgr5 promotes the formation of cytonemes in mammalian cells suggesting a possible role for cytonemes in gastrointestinal cancer cell survival, invasion, and metastasis [113]. Exo70, a key component of the Exocyst complex, induces extensive actin membrane protrusions resembling filopodia and blocking Exo70 function inhibits invadopodia formation [114]. *Exo70* expression is upregulated in colon cancer samples and its expression is positively correlated with tumor size, invasion depth, and distant metastasis. Colon cancer patients with higher *Exo70* expression have a poorer clinical outcome than those with lower *Exo70* expression [115].

In particular, glioblastoma (GB) cells produce long cellular protrusions at the invasive edge of the tumor that scan the surrounding area and interconnect tumor cells. These protrusions are F-actin based and form a complex network that interconnects GB cells; therefore, they are named tumor microtubes (TMs) [116]. TMs contribute to invasion and proliferation, resulting in effective brain colonization by GB cells. Moreover, TMs constitute a multicellular network that connects GB cells over long distances (up to 500 µm length) [116]. This study found that Growth Associated Protein-43 (GAP43) is essential for the development of TMs and the tumor cell network associated with GB progression, and it drives TM-dependent tumor cell invasion, proliferation, interconnection, and radioresistance. TMs share many characteristics with cytonemes, they are actin-based projections and they are marked by several cytoneme markers, including Ihog, LifeActin, GMA, GPI, myosin light chain (MLC), and the nonmuscle type 2 myosin, spaghetti squash (sqh). Moreover, this study [86] showed in a *Drosophila* glioma model that the glioma network is dependent on the fly *GAP43-like* gene (*igloo*, *igl*), as has been described in human tumor cells. The glioma network does not develop upon *igl* silencing. TMs stability in GB is sensitive to *GAP43* expression in tumoral cells. Moreover, downregulating *Nrg* (*Nrg-RNAi*), which is known to prevent epithelial cytoneme formation, resulted in a reduction of the TM-like processes in GB [86]. Moreover, TMs accumulate Frizzled1 receptor (Fz1) that mediates Wg signaling (Figure 3) [86]. Thus, there are molecular and functional similarities between cytonemes and TMs; however, the term cytoneme is used for physiological situations, and TMs is restricted to the tumoral condition.

TMs and TNTs share some structural features, but TMs are more stable, longer, and thicker (2 µm). In addition, TMs in human cells provide functional coordination to GB cells and facilitate cell repair, brain infiltration, and offer resistance to radiotherapy through dilution of Ca^+2^ intracellular peaks [116], which thereby increases the aggressiveness of GB.

## 5. Concluding Remarks

Filopodia are a cellular system of communication widely expanded among living organisms from bacteria to human cells. These protrusions mediate the interaction among cells and with their microenvironment and serve as sensors for the filopodia-emitting cells. Over decades, different forms of filopodia have been described according to their specific function, composition, dimensions, and stability.

Among them, cytonemes have emerged as a novel alternative for cell to cell communication that are involved in development, physiology, and disease. Cytonemes contribute to the directionality of the signals and the specificity of the interaction, as there are emitter and receiver cytonemes for specific signaling pathways. In particular, Wg/WNT, Dpp/BMP, and Hh signaling can be mediated by cytonemes during development and are essential for certain tumoral cell types progression. This feature brings a novel perspective for cancer biology and reveals potential targets for treatment. Therefore, there is a need to decipher the specific mechanisms underlying cytoneme formation, and in general, each type of filopodial protrusion.

Again, the discoveries from animal models, such as *Drosophila*, provide novel approaches to understand the role of cytonemes in central processes in biology and how they are involved in tumorigenesis. Since cytonemes are conserved structures in other animals, including human cells, the challenge for scientific research in the following years will be to understand the molecular basis of their function in normal physiology and cancer.

## Figures and Tables

**Figure 1 ijms-20-05641-f001:**
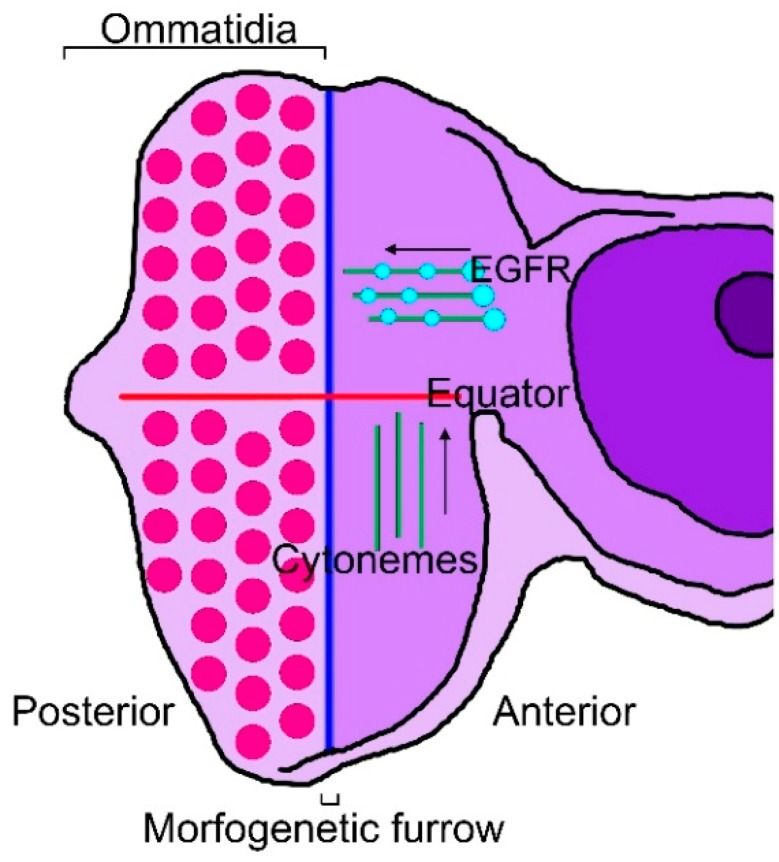
Cytonemes of the eye imaginal disc. Diagram of an eye disc in which the MF (blue line) has progressed from posterior to anterior, showing differentiated ommatidia in the posterior region of the eye disc (red dots). The equator is perpendicular to the MF (red line). There are two types of cytoneme (green lines) extending from cells anterior to the MF: (1) cytonemes oriented toward the equator and (2) cytonemes oriented toward the MF and populated with EGFR-containing puncta (light blue dots).

**Figure 2 ijms-20-05641-f002:**
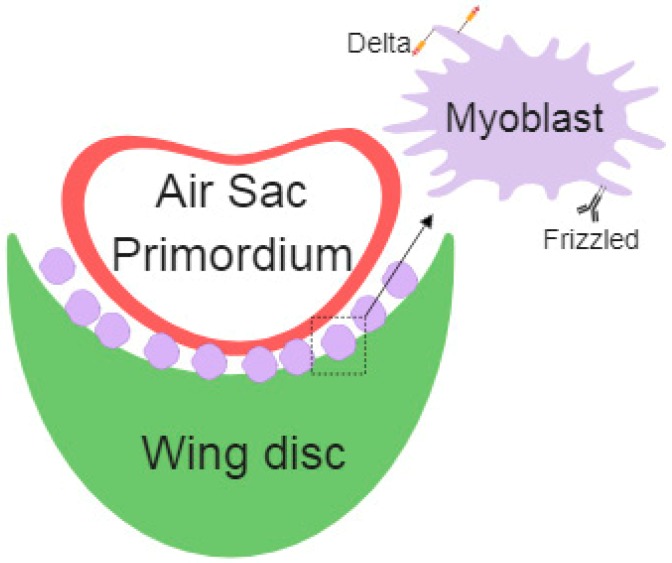
Schematic representation of air sac primordium (ASP), wing disc epithelial cells, and myoblasts cells with specific Delta- or Frizzled-cytonemes.

**Figure 3 ijms-20-05641-f003:**
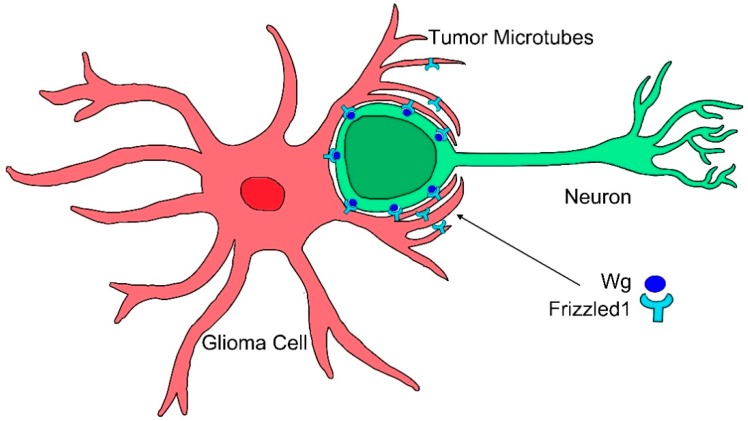
Cytonemes in tumorigenesis. Glioma cells produce a network of tumor microtubes that grow to reach and enwrap neighboring neurons. Increased glia-neuron membrane contacts facilitate neuronal Wg sequestering mediated by glioma Frizzled1 receptor accumulated in the tumor microtubes.

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
