# Peer review of "Cytonemes, Their Formation, Regulation, and Roles in Signaling and Communication in Tumorigenesis"

_ijms, 2019, doi:10.3390/ijms20225641_

Round 1

Reviewer 1 Report

Yhe manuscript is very interesting and well written and well structured.
I ask the authors to better explain the role of Cytonemes in tumors and whether there are studies about cancer colon

Author Response

Response to Reviewer 1 Comments

Point 1: Yhe manuscript is very interesting and well written and well structured.
I ask the authors to better explain the role of Cytonemes in tumors and whether there are studies about cancer colon

Response 1: We have included further explanation on the role of cytonemes in tumor progression and discussed the latest literature about cytonemes and colon cancer.

Reviewer 2 Report

This review on the cytosome was interesting and well written. The only issue I found with it is the general organization of the paragraphs. 

First the authors have a section on "Cytonemes in Physiology of Epithelial cells". This section is overwhelmingly on Drosophila development. This should probably be reflected in the title.

Then they have a section on "Cytonemes in Non-Epithelial Tissues: Trachea, Myoblast, Embryo, Ovary, Brain". Within this section, the sub-section "3.1. Trachea", focuses mostly on the role of cytonemes in neurons and the relationship with glutamate signaling, so this does not belong in the "Trachea" section. Then there is a sub-section "3.2. Myoblasts", that still talks about the air sac, so its is still related to the tracheal development and should probably be in the "Trachea" section. The sub-section "3.4. Embryo" summarizes concepts that have already been presented, because most of the previous sections were on Drosophila development, so it is not clear why there shuold be a sub-section dedicated to "embryo" in general.

Finally, the section "4. Cytonemes in Pathology: Tumourigenesis" presents the role of tunneling nanotubes in a Drosophila tumor and in glioblastoma. This is an interesting section, but the glioblastoma is a special tumor that derives from the nervous system. Can the authors find and discuss examples of nanotubes in epithelial tumors?

Author Response

Response to Reviewer 2 Comments

Point 1: This review on the cytosome was interesting and well written. The only issue I found with it is the general organization of the paragraphs. 

First the authors have a section on "Cytonemes in Physiology of Epithelial cells". This section is overwhelmingly on Drosophila development. This should probably be reflected in the title.

Response 1: We have modified the title of the section and included an explanatory sentence accordingly for a better understanding.

“In this section we will review and summarize the available information about the structure and composition of cytonemes, available markers, and components required for cytoneme formation, most of the literature available is based in the Drosophila melanogaster model. Later we will review cytoneme physiological roles in cell-cell signaling in both epithelial and in non-epithelial cells.”

Point 2: Then they have a section on "Cytonemes in Non-Epithelial Tissues: Trachea, Myoblast, Embryo, Ovary, Brain". Within this section, the sub-section "3.1. Trachea", focuses mostly on the role of cytonemes in neurons and the relationship with glutamate signaling, so this does not belong in the "Trachea" section.

Response 2: Recent publications describe the presence of canonical glutamate synapse components in the Air Sac Primordium (ASP). Thus, we have clarified the terms of neuronal synapses and cytoneme synapses, and defined cytoneme synapses at the beginning of the section.

Point 3: Then there is a sub-section "3.2. Myoblasts", that still talks about the air sac, so its is still related to the tracheal development and should probably be in the "Trachea" section.

Response 3: We have modified this section to differentiate between the trachea and the myoblast sections. We would like to point out that the trachea sub-section is about cytonemes emerging from the air sac cells to contact epithelial cells in the wing disc, on the other hand, the Myoblast section is about cytonemes emerging from the myoblasts to contact either epithelial wing cells for Wg signalling or ASP cells for Notch signalling.

Point 4: The sub-section "3.4. Embryo" summarizes concepts that have already been presented, because most of the previous sections were on Drosophila development, so it is not clear why there shuold be a sub-section dedicated to "embryo" in general.

Response 4: We fully agree with the reviewer, therefore we have included the studies in the chick embryo in the introduction and deleted the embryo section.

Point 5: Finally, the section "4. Cytonemes in Pathology: Tumourigenesis" presents the role of tunneling nanotubes in a Drosophila tumor and in glioblastoma. This is an interesting section, but the glioblastoma is a special tumor that derives from the nervous system. Can the authors find and discuss examples of nanotubes in epithelial tumors?

Response 5: We discuss the first 2 studies about the role of signalling through cytonemes in tumours and we have included a section dedicated to this issue.